# Estimation of Sugar Content in Wine Grapes via In Situ VNIR–SWIR Point Spectroscopy Using Explainable Artificial Intelligence Techniques

**DOI:** 10.3390/s23031065

**Published:** 2023-01-17

**Authors:** Eleni Kalopesa, Konstantinos Karyotis, Nikolaos Tziolas, Nikolaos Tsakiridis, Nikiforos Samarinas, George Zalidis

**Affiliations:** 1Laboratory of Remote Sensing, Spectroscopy, and GIS, School of Agriculture, Aristotle University of Thessaloniki, 57001 Thermi, Greece; 2School of Science and Technology, International Hellenic University, 14th km Thessaloniki—N. Moudania, 57001 Thermi, Greece

**Keywords:** TSS, vis–NIR, NIR spectroscopy, cultivar, vineyard, deep learning, oenological parameters

## Abstract

Spectroscopy is a widely used technique that can contribute to food quality assessment in a simple and inexpensive way. Especially in grape production, the visible and near infrared (VNIR) and the short-wave infrared (SWIR) regions are of great interest, and they may be utilized for both fruit monitoring and quality control at all stages of maturity. The aim of this work was the quantitative estimation of the wine grape ripeness, for four different grape varieties, by using a highly accurate contact probe spectrometer that covers the entire VNIR–SWIR spectrum (350–2500 nm). The four varieties under examination were Chardonnay, Malagouzia, Sauvignon-Blanc, and Syrah and all the samples were collected over the 2020 and 2021 harvest and pre-harvest phenological stages (corresponding to stages 81 through 89 of the BBCH scale) from the vineyard of Ktima Gerovassiliou located in Northern Greece. All measurements were performed in situ and a refractometer was used to measure the total soluble solids content (°Brix) of the grapes, providing the ground truth data. After the development of the grape spectra library, four different machine learning algorithms, namely Partial Least Squares regression (PLS), Random Forest regression, Support Vector Regression (SVR), and Convolutional Neural Networks (CNN), coupled with several pre-treatment methods were applied for the prediction of the °Brix content from the VNIR–SWIR hyperspectral data. The performance of the different models was evaluated using a cross-validation strategy with three metrics, namely the coefficient of the determination ($R^{2}$), the root mean square error (RMSE), and the ratio of performance to interquartile distance (RPIQ). High accuracy was achieved for Malagouzia, Sauvignon-Blanc, and Syrah from the best models developed using the CNN learning algorithm ($R^{2}>0.8$, RPIQ≥4), while a good fit was attained for the Chardonnay variety from SVR ($R^{2}=0.63$, RMSE=2.10, RPIQ=2.24), proving that by using a portable spectrometer the in situ estimation of the wine grape maturity could be provided. The proposed methodology could be a valuable tool for wine producers making real-time decisions on harvest time and with a non-destructive way.

## 1. Introduction

Grapes are the fifth most widely produced fruit, with approximately 79.5 million tonnes being produced worldwide in 2019 [1]. The estimation of the maturity of the berries is critical for determining the appropriate harvesting time, which in turn results in favorable quality [2]. This is particularly important in wine production, where the optimal harvest time determined by enological parameters largely affects the produced wine quality. The exact point in time in which each grape berry attains its optimal maturity level depends, among others, on the terrain [3], the duration of the growing season, the variety, the vine tree load, and other environmental conditions [4,5].

The common practice used today by the wine producers to assess the maturity of the grapes of a field is a two-step process. First, experts use sensory attributes (i.e., color and taste) to identify that a grape is entering its maturity stage. Following this, they sample some of the fruits, crush them, and use a refractometer to measure their °Brix value in situ [6]. A similar step may be performed in the laboratory using more costly chemical analyses. The °Brix value is the total soluble solids content, i.e., the sugar content of the aqueous solution of the crushed grape. The sugar content is crucial since it determines the alcoholic level of the wine and is critical in the subsequent fermentation process [7]. The decision about when to harvest is currently taken based on this sampling method, which relies on the destructive measurements at a given time, without considering the evolution of the associated variables. Evidently, this traditional approach cannot reliably perform selective harvesting [2].

On the other hand, near infrared (NIR) reflectance spectroscopy is a widespread technique used in many scientific fields such as in the food quality assessment sector [8] and soil science [9]. It is a non-destructive technique based on measuring the electromagnetic radiation, which is reflected at different wavelengths by the target surface, and may be used for the quality evaluation of intact fruits and vegetables [10]. In the context of grape composition monitoring during the ripening process using NIR spectroscopy, a variety of works has been published [11,12,13,14,15,16], but most of these were conducted under controlled conditions, such as illumination, temperature, humidity, and sample positioning, among others—i.e., in a laboratory. For example, in a recent study [17], three ripening parameters were predicted in four table grape cultivars using a NIR sensor under laboratory conditions. Chariskou et al. [18] used Fourier-transform near infrared reflectance spectra of intact berries to predict total soluble solids (TSS) content, but the spectra were recorded in the laboratory. Line-scan hyperspectral imaging in the NIR has also been employed in laboratory conditions to estimate sugar and pH levels in wine grape berries [19]. Other studies (e.g., Ref. [20]) have focused on studying the extracted berry juice, which may potentially yield better results but requires a destructive measurement process. Larrain et al. [21] developed a portable instrument for determining the ripeness in wine grapes using NIR spectroscopy (640–1100 nm), testing it in five grape varieties in Chile, but used only a fraction of the NIR spectrum.

The use of hyperspectral data and specifically in the visible and near infrared (VNIR, 350–1000 nm) and in the short-wave infrared (SWIR, 1000–2500 nm) regions in grape production is of great interest [22]. Due to its fine spectral resolution, hyperspectral analysis can contribute to both fruit monitoring and quality control at all stages of maturity in a simple and inexpensive way. The regions of VNIR and SWIR in grape production have piqued the interest of researchers in the past [23]. Specifically, three comprehensive reviews show the potential and challenges of infrared spectroscopy for analyzing the chemical composition of grapes in the laboratory, the vineyard, and before or during the harvest, to provide better insights into the chemistry, nutrition, and physiology of grapes [24,25,26].

Furthermore, machine learning has been applied in the wine sector across various domains and the entire production chain. In addition to providing estimations of grape maturity indicators, a plethora of other applications has been reported in the literature. For example, machine learning approaches can assist in determining how consumers perceive wine quality, thus constituting an effective tool for understanding the complex nature of wine datasets and conveying useful information related to wine quality [27]. Machine learning coupled with various spectroscopic approaches has also been employed to guarantee wine authenticity [28]. Moreover, wine grape yield was estimated at the regional level in Portugal using as inputs a combination of space-borne imagery and climate data in a deep learning framework [29]. Another interesting application involves the segmentation of the grapevines and the vineyard classification, which are key to optimize the management of vineyard plots; this was achieved through the use of aerial imagery obtained from unmanned aerial vehicles [30]. A machine learning method has also been developed to predict which municipalities will obtain geographical Indications in the future, showcasing which main features (e.g., territorial conditions, socio-economic factors, etc.) are more relevant in predicting their success [31]. These works, although not presenting exhaustively all possible applications of machine learning in the wine sector, demonstrate the merits of the data-driven approaches, which can improve the performance on some tasks or provide essential tools for various stakeholders.

The main goals of the current work were to (i) create a local grape spectral library and (ii) estimate the wine grape ripeness in a simple, inexpensive, and non-destructive way, taking advantage of the synergistic use of ground truth °Brix measurements, in situ point spectroscopy data, and machine learning algorithms coupled with several pre-processing techniques.

## 2. Materials and Methods

### 2.1. Materials

#### Study Area

The data were collected during the harvesting seasons of 2020 and 2021 (early July to late August) from the Ktima Gerovassiliou vineyard, located on the slopes of Epanomi in Northern Greece, 25 km southeast of the city of Thessaloniki (40°27′04″ N, 22°55′23″ E). The climate in the region of Epanomi is Mediterranean with mild winters and hot summers, tempered by sea breezes. The soil is mainly sandy with a few clayey substrates and calcareous rocks; it is rich in sea fossils since the surrounding hilly area was formed by sea deposits.

Ktima Gerovassiliou is one of the largest wine producers in the country, cultivating several varieties of both white and red wine. It is a single private vineyard stretching over 72 hectares containing about 400,000 vine trees (Figure 1). In this work, we examined the following grape varieties: Chardonnay, Malagouzia, Sauvignon-Blanc, and Syrah; they were selected considering that Malagouzia is a local Greek wine grape while the rest are among the most popular wine varieties worldwide. The vine trees used in the study were planted in 2017 using a conventional agricultural system with a high wire cordon trellis system while drip irrigation is employed.

### 2.2. Methods

A graphical representation of the experimental procedure adopted in the current work is provided in Figure 2. In essence, the sampling procedure with both a refractometer and a spectroscopy sensor in the four different grape varieties was the initial step from where the ground truth data (including the grape spectral library) were collected. After the establishment of the grape spectral library, the regression models were developed. It is important to note that in the modeling stage, we conducted our analysis for each grape variety independently. By using the 5-fold cross-validation (CV) technique to split the dataset to training and test sets and through the application of machine learning algorithms, the final sugar content values could be estimated from the spectral data.

#### 2.2.1. Equipment, Preparation, and Protocols

As part of the work and for the development of the grape spectral library, an appropriate preparation was made and specific protocols for collecting the measurements were defined. More specifically, in the initial stage, the study area was prepared, and labels were placed in different bunches of the varieties, so that the repetitions of the measurements were as representative as possible in order to have the maturity level in different phenological stages. Specifically, we placed (i) 20 in Chardonnay variety (ii) 30 labels in Malagouzia variety, (iii) 20 in Sauvignon-Blanc, and (iv) 20 in Syrah variety. This significantly organized the vineyard area and sped up the measurement process in an accurate and precise way. Because the Malagouzia variety was not measured in the summer of 2020, it was considered appropriate to include 10 more bunches than the other varieties during the summer of 2021. Figure 3 presents the four different grape varieties with their labels in the field.

Spectral acquisitions were performed using a portable contact probe spectrometer to get the reflectance spectra of the grapes. In this regard, all measurements were performed in situ using the PSR+3500 (Spectral Evolution Inc., Lawrence, MA, USA) spectrometer that covers the VNIR–SWIR spectrum (350–2500 nm). The spectrometer uses one 512-element Si photodiode array for the 350–1000 nm range with a full width at half maximum (FWHM) resolution of 2.8 nm at 700 nm, a 256-element InGaAs detector covering the 970–1910 nm range with a FWHM resolution of 8 nm at 1500 nm, and finally a second 256-element InGaAs photodiode array for the 1900–2500 nm range having a FWHM resoltion of 6 nm at 2100 nm. The contact probe of the spectrometer came into contact with the berry in the cluster to record its spectrum. The effect of ambient light was reduced by ensuring that the sample was in the shade (natural or artificial). The spectra were acquired in reflectance mode by recording the average of five measurements, after the necessary calibration process by using a white reference plate made of Spectralon® material. This calibration procedure was repeated after 10 successful spectra measurements in order to keep the sensor accurate and calibrated.

In addition, we recorded the °Brix content of the same grapes measured with the spectrometer with the help of a portable RHB-32ATC refractometer (Laxco Inc., Bothell, WD, USA), but with a destructive effect of the fruit. The refractometer has a range of 0–32 °Brix, an accuracy of 0.20 °Brix, and a resolution of ±0.20 °Brix. In the initial stage and before each new measurement the prism of the refractometer was rinsed with deionized water and dried with absorbent cleaning paper. The refractometer was calibrated by using a drop of distilled water (at about 20 °C) and by adjusting the instrument to read zero degrees of °Brix. The grapes measured with the spectrometer were carefully cut and squeezed to use their juice for analysis and placed on the transparent prism surface of the refractometer. As the scale for measuring the °Brix through the lens is not visually legible, the samples were checked independently by two different agronomists and the average of the two was taken as the final °Brix value. After completing the measurements for the sampling day, the final °Brix values were also checked by the agronomist’s expert opinion of Ktima Gerovassiliou in order to validate our results.

#### 2.2.2. Grape Spectra Library Development—Field Sampling

A total of four varieties were selected for sampling and specifically, the grape sampling was carried out during the pre-harvest and harvest period (July to August) for the years 2020 and 2021, in order to have representative measurements in the total range of grape ripeness. According to the BBCH scale [32] which describes the phenological growth stages of the grapevine, we monitored stages 81 (beginning of ripening where the berries begin to brighten in color) to 89 (berries are ripe for harvest). Due to the rapid change in the grapes’ sugar content, the measurements were collected on a weekly basis and in that way, different stages of maturity were considered. In addition, taking into consideration the influence of environmental conditions, especially the high air temperatures and direct sunlight, the measurements were consistently taken early in the morning. The measurements were performed per bunch where berries of different sizes were chosen, in order to have sugar content variability depending on the period.

#### 2.2.3. Spectroscopy Pre-Treatments

Data pre-processing is an important step in chemometrics analysis and often has a significant impact on the generalization performance of a supervised machine learning algorithm. The object of the spectral pre-treatments is to remove physical phenomena in the spectra to improve the subsequent modeling phase [33]. The most widely used pre-processing techniques, also used in this paper, are divided into two categories: scatter-correction methods and spectral derivatives. Initially, the reflection data recorded were converted to pseudo-absorbance, which may present a more linear relationship between absorption and concentration of the grape properties according to the empirical Beer–Lambert law. Furthermore, the method of the first derivative is a method that removes the baseline from the spectra and at the same time emphasizes the absorption characteristics. According to Ertlen et al. [34] important information can be extracted from the VNIR spectra if spectral derivatives are obtained. A Savitzky–Golay 1st order filter (SG1) is usually employed to calculate the first derivative [35]. As a first step, we performed downsampling on the initial recorded reflectance spectra. The data output of the instrument uses a step of 1 nm, thus generating 2151 distinct data points. However, the true spectral resolution (full width at half maximum) of the instrument is coarser, at about 2.8 nm at 700 nm, 8 nm at 1500 nm, and 6 nm at 2100 nm. We accordingly performed downsampling at a 10 nm resolution to reduce the feature space. Thus, to be precise, the number of input features considered in the modeling process is 216 with the set of input wavelengths being {350+10k∣k∈{0,1,…,215}} (expressed in nm). In addition, we applied the following common spectral pre-treatments to the original reflectance spectra:The standard normal variate (Ref+SNV);A first order Savitzky–Golay filter of a window length of 5 and a polynomial order of 3 (Ref+SG1);The pseudo-absorbance transformation, calculated as –log10(R) and denoted as Abs;The standard normal variate after the pseudo absorbance transformation (Abs+SNV);A first order Savitzky–Golay filter of a window length of 5 and a polynomial order of 3 after the pseudo absorbance transformation (Abs+SG1);The previous step followed by the standard normal variate (Abs+SG1+SNV);A second order Savitzky–Golay filter of a window length of 5 and a polynomial order of 3 on the pseudo absorbance spectra, followed by the standard normal variate (Abs+SG2+SNV); andThe continuum removal transformation (CR).

Thus, with the eight aforementioned pre-treatments and the initial reflectance spectra, in total we tested nine different spectral inputs.

#### 2.2.4. Models

To estimate the sugar content from the spectra, the following standard modeling algorithms were applied:Partial Least Squares regression (PLS) [36]: PLS has only one hyperparameter, namely the number of latent variables; to optimize it we searched within [1, 100];Random Forest (RF) [37]: To optimize the hyperparameters of RF, a grid search was conducted as follows: the number of trees was selected from the 50, 100, 150, 200 set while the maximum number of features to consider when looking for the best split were selected from the {“max”, “sqrt”, “log2”} set;Support Vector Regressor (SVR) [38]: In addition to selecting the RBF kernel, the model was optimized through a grid search for its hyperparameters by examining the following values for ϵ: {0.01, 0.025, 0.05, 0.075, 0.10, 0.15, 0.20}, while the cost C was selected from $\left\{2^{-2},2^{-1},\ldots ,2^{9}\right\}$.

Additionally, we implemented a custom CNN, which is based on the one proposed by Tsakiridis et al. [39]. Briefly, this deep learning network comprises one-dimensional convolutional layers used as feature extractors followed by fully connected layers to provide the inference (prediction value). To optimize both the model (i.e., structural) and the learning algorithm’s hyperparameters, and considering that a grid search which examines all possible combinations is expensive for deep learning algorithms, we performed tuning using the Hyperband optimization algorithm [40]. The various hyperparameters are presented in Table 1. Input standardization refers to the pre-processing of the input matrix, which may be none, per-band standardization between −1 and 1, and using the standard score (i.e., removal of mean and standard deviation of 1). The only hyperparameters that were static (i.e., not optimized via the Hyperband algorithm) are:The maximum training epochs which were set to 200;The use of an early stopping mechanism, which halts the training process when the monitored metric of the loss in the validation set has stopped improving, with a patience of 40 (i.e., the number of epochs with no improvement after which training will be stopped);A learning rate scheduler with a step decay schedule which halves the learning rate every 60 epochs;The dense layers use kernel regularization with an L2 regularization penalty of 0.0004.

#### 2.2.5. Dataset Split

The dataset is usually divided into training and test sets for modeling purposes; the first is used in order to train and optimize the model’s hyperparameters (model selection) while the latter is to estimate the model prediction error or accuracy [41]. This split was done separately for every grape spectrum dataset per variety. Instead of a single split between training and test sets, we employed a 5-fold cross validation approach to evaluate the models’ performance more robustly. Figure 4 presents the kernel estimation plots for the five different folds used per each variety, to identify if there are important differences in the distribution of the output property across the different folds used. Thus, the model development phase was repeated five times. At each iteration, each model’s hyperparameters were optimized with another (internal) 5-fold cross validation approach, as presented in Algorithm 1. The procedure for the CNN model is similar, albeit instead of a complete grid search, the Hyperband optimization algorithm is used (Algorithm 2).
Figure 4A kernel density estimation plot for the five different folds used per each variety.
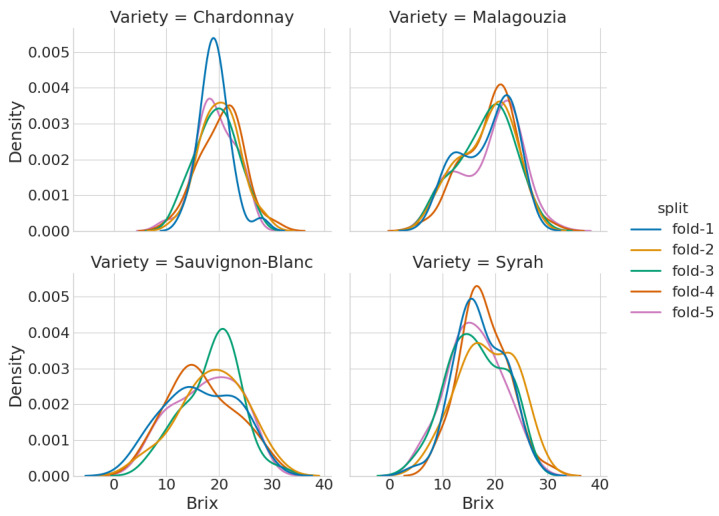

**Algorithm 1:** Model training and hyperparameter optimization for the standard AI models.**Require:** Sets of model hyperparameter values to evaluate      ▹ Cartesian product**Ensure:** Average model performance across all folds1:**for** each external fold iteration **do**                ▹ External 5-fold CV2:    Hold-out the samples of this fold (i.e., the test set)3:    Use the rest of samples as calibration set4:    **for** each hyperparameter set **do**5:        **for** each internal fold iteration in the calibration set **do**    ▹ Internal 5-fold CV6:           Hold-out the samples of this fold (i.e., the validation set)7:           Fit the model on the rest (i.e., the training set)8:           Use the model to predict the held-out samples9:           Calculate and store an accuracy metric for this fold10:        **end for**11:        Calculate the average performance across all folds12:    **end for**13:    Determine the optimal hyperparameter set14:    Fit the final model using the calibration set with the optimal hyperparameters15:    Use the model to predict the held-out samples16:    Calculate and store accuracy metrics for this fold17:**end for**18:Calculate the average performance across all folds

**Algorithm 2:** Model training and hyperparameter optimization for the CNN model.
1:Define ranges of model hyperparameter values to evaluate2:**for** each external fold iteration **do**              ▹ External 5-fold CV3:    Hold-out the samples of this fold (i.e., the test set)4:    Use the rest of samples as calibration set5:    **for** each iteration of the Hyperband algorithm **do**6:        **for** each internal fold iteration in the calibration set **do**  ▹ Internal 5-fold CV7:           Hold-out the samples of this fold (i.e., the validation set)8:           Fit the model on the rest (i.e., the training set)9:           Use the model to predict the held-out samples10:           Calculate and store an accuracy metric for this fold11:        **end for**12:        Calculate the average performance across all folds13:    **end for**14:    Determine the optimal hyperparameter set15:    Fit the final model using the calibration set with the optimal hyperparameters16:    Use the model to predict the held-out samples17:    Calculate and store accuracy metrics for this fold18:
**end for**
19:Calculate the average performance across all folds


#### 2.2.6. Model Evaluation Metrics

The models were validated on the independent test set using the following metrics:The coefficient of determination $R^{2}$;The root mean squared error (RMSE); andThe ratio of performance to interquartile range (RPIQ).

$R^{2}$ quantifies the degree of any linear correlation between the observed and the model predicted output; it usually ranges from 0 to 1 (higher is better) and is calculated as: (1)$$R^{2}\left(y,\hat{y}\right)=1-\frac{\sum_{i=1}^{N}{\left(y_{i}-{\hat{y}}_{i}\right)}^{2}}{\sum_{i=1}^{N}{\left(y_{i}-\bar{y}\right)}^{2}}$$
with ${\hat{y}}_{i}$ being the prediction for the *i*-th pattern, $y_{i}$ its ground truth value, and $\bar{y}$ the mean ground truth value across all *N* patterns.

RMSE is calculated via: (2)$$\mathrm{RMSE}\left(y,\hat{y}\right)=\sqrt{\frac{\sum_{i=1}^{N}\left(y_{i}-{\hat{y}}_{i}\right)}{N}}$$

RPIQ on the other hand takes both the prediction error and the variation of observed values into account, without making assumptions about the distribution of the observed values. It is defined as the interquartile range of the observed values divided by the RMSE of prediction [42]: (3)$$\mathrm{RPIQ}\left(y,\hat{y}\right)=\frac{Q_{3}-Q_{1}}{\mathrm{RMSE}(y,\hat{y})}$$
where $Q_{1}$ is the lower quartile or the 25th percentile of the data, whereas $Q_{3}$ is the upper quartile, which corresponds with the 75th percentile.

#### 2.2.7. Interpretability Analysis

We applied an interpetability analysis on two levels: pre-hoc (i.e., before model development) and post-hoc (i.e., after model development). With respect to the first level, the mutual information [43] between the input (spectral data) and the output (°Brix) was calculated in the original reflectance space for each grape variety. The mutual information between two random variables is a non-negative value, quantifying the dependency between the variables; a zero value indicates that the two random variables are independent, while higher values suggest a higher dependency between them.

To understand how each model identifies the association between the input and the output, namely between the spectra and the sugar content measured in degrees °Brix, a post-hoc interpretability analysis was conducted. That is, after each model was trained, we applied model-specific techniques to elicit information to shed light on how the input–output relationship is identified. The techniques examined in this paper are tailored or specifically designed to explain the corresponding artificial intelligence (AI) models, and are iterated below per each model:PLS: One of the most popular techniques is to apply the variable influence on the projection (VIP) method, also known as variable importance in projection or VIP scores [44]. The calculation of the VIP scores, which assign a value to each input feature in the initial space, considers the amount of explained variance of the output in each extracted latent variable. They give a measure useful to identify which of the input features contribute the most to the output variance explanation. In essence, they can be utilized to perform feature ranking; or in the particular case examined herein to identify which input wavelengths are associated the most with the sugar content. The average of the squared VIP scores is equal to 1, and sometimes the greater-than-one rule is used to identify the most important features, which is not a statistically justified limit but can be considered as a rule-of-thumb;RF: Tree ensembles combine different trees to obtain an aggregated regression result, which—although effective against overfitting—has an adverse effect: the increased complexity in the interpretation of the overall ensemble over that of each of its compounding tree learners. The Gini importance, known also as the impurity-based feature importance [45], was calculated for the RF models. It is computed as the (normalized) total reduction of the criterion brought by that feature;SVR and CNN: Both these models are considered black-box models [46] whose interpretability degree is low. Nevertheless, model agnostic techniques may be applied to calculate the feature importance [47]. In this paper, we applied SAGE (Shapley Additive Global importancE) [48], which is a model agnostic game-theoretic approach summarizing each feature’s importance based on the predictive power it contributes, while it also accounts for complex feature interactions using the Shapley value [49].

It should be noted that each of these methods was applied across the five folds, and a mean feature ranking was calculated. This may produce less sparse results; for example, adjacent wavelengths may be identified as important in different folds. Nevertheless, this also has the benefit of identifying more robustly which features are regarded as important as they should be dominant across all folds.

## 3. Results

### 3.1. Dataset

An overall description of the dataset descriptive statistics obtained by conventional techniques for grape sugar content for all the varieties is presented in Table 2. These values were used as ground truth data to test the machine learning prediction models. Based on the table it is possible to observe differences between varieties. Furthermore, the minimum and maximum values for all varieties have significant ranges due to the measurements that took place from the early ripening stage of the grapes in July until just before their harvest time in August. Overall, within the varieties, values range from 4.2 up to 31.4 °Brix, where both the extreme values are found in the Sauvignon-Blanc variety. Regarding the mean standard deviation and the quartiles, it should be noted that they vary widely among the varieties. In addition, for more information about the statistics of the grape dataset, the box plot of Figure 5 for the sugar content ground truth values is provided.

### 3.2. Spectra Analysis

Figure 6 illustrates the reflectance spectra of the grape spectra library for the four varieties. The x axis defines the wavelength (350 to 2500 nm) and the y axis the reflectance. The continuous line is the mean spectrum, while the shaded area indicates the confidence interval calculated at a 95% confidence level. Looking carefully in the graph it is obvious that the white grape varieties (Chardonnay, Malagouzia, Sauvignon-Blanc) present the same variations in contrast to the red variety (Syrah), which presents certain differences, especially in the range of 450 to 700 nm.

How the grapes’ spectral signatures are affected as they grow more mature (i.e., with higher °Brix content) is shown in Figure 7. An important point that arises from this figure is that the valuable information pertaining to the maturity of the grapes according to all spectral signatures focuses on the range from 550 to 1300 nm, whereas most of the SWIR appears to provide limited information in terms of absorption bands. At least in terms of the mean spectra, it is possible to visually identify some patterns that indicate a transition from low to higher sugar content; e.g., for Chardonnay and Malagouzia at 680 nm and for Syrah from 700 to 900 nm. Things are more convoluted for Sauvignon-Blanc, where the mean reflectance curve for the higher sugar content is found in between the respective curves for low and medium content. Yet even in this case, there is a visible absorption band at approximately 750 nm that increases for higher sugar content values. This indicates that by using spectral pre-processing techniques it may be possible for an apt learning algorithm to discern a more robust pattern and infer the sugar content from the spectral data.

The effect of the various spectral pre-treatments is presented in Figure 8. The absorbance and continuum removal transformations highlight five main absorbance regions, namely at around 680, 970, 1200, 1440, and 1920 nm, which are dominant throughout all grape varieties and irrespective of the grapes’ sugar content. The SNV transformation, which is a scatter-correction technique, also adjusts for baseline shifts between samples. As expected, the effect on the mean spectra is not profound, however, when comparing, e.g., the Ref and Ref+SNV spectra, the standard deviation indicates that the largest deviations are to be found in the VNIR. On the other hand, the first-derivative transformations indicate the same sharp peaks, albeit with a small offset, which is expected due to the mathematical calculation of the first-derivative; it is zero exactly at the peak and attains its largest positive and negative value at each side. Therefore, care should be taken when reading into the exact wavelengths from the first-derivative transformation. For similar reasons, the second-derivative transformation does not have the same problem and its peaks indicate more precisely the exact location of the absorption bands.

### 3.3. Prediction Accuracy

The mean prediction of accuracy across the five-folds in the independent test set is given in Table 3. Therein we present for each model the best spectral pre-treatment, which was determined as the one that minimized the mean RMSE in the internal validation sets. To aid the reader in the comparison, Figure 9 provides a visual comparison of the RPIQ values. The optimal set of hyperparameters corresponding to those results is provided in Table A1. Additionally provided in Figure 10 are the predictions of the CNN model using the Ref+SG1 spectra for the Syrah variety in the independent test set (i.e., the best model); the respective results of Table 3 are the mean metrics calculated across the five-folds. It is thus visible that there are some slight variations across the folds, which justifies our selection to use a five-fold cross-validation strategy to provide more robust comparisons.

The Chardonnay variety yielded the worst results across all four different models, and only the CNN and SVR models attained a performance of mean RPIQ > 2 and mean $R^{2}$ >= 0.6, which indicates a good fit. In the other three varieties, namely Malagouzia, Sauvignon-Blanc, and Syrah, the CNN model attained the best results when compared with the other models. It should be noted that in these varieties the mean RPIQ was greater than 4 while the $R^{2}$ metric was greater than 0.8, suggesting that the prediction accuracy was very good.

When the different learning algorithms are compared, it is evident that the CNN model achieved the second best performance in the Chardonnay variety (behind SVR) and the best results in the remaining three out of four varieties. In particular, for Chardonnay, the relative difference in terms of RMSE is about 4.6%, which expressed in °Brix is about 0.1. The greatest difference is observed for Sauvignon-Blanc, where the CNN model compared to the second best (namely, SVR), has a relative difference in terms of RMSE of about 20%, or 0.5 °Brix. It should also be noted that the differences in Malagouzia across all models are quite small, and all four algorithms attained a very similar performance. Finally, the smallest prediction error was to be found in the only red wine variety in our study, namely Syrah, where the CNN model had an RMSE of 1.76 °Brix.

With respect to the best pre-treatment, it must be highlighted that even though it was not universal across all learning algorithms and varieties, nevertheless there was a strong preference for the use of the 1st derivative spectra, with variations including it producing the best results in 14 out of 16 cases. This attests to the fact that by accentuating the absorption bands it is possible to yield lower prediction errors. Most notably, in both cases where the 1st derivative spectra were not the most favorable approach, the PLS algorithm was the model used. In the first one (i.e., PLS in Chardonnay) the model had a poor fit and used the 2nd derivative spectra, while in the second case (i.e., PLS in Malagouzia) the absorbance spectra case were utilized with a noteworthy performance, very close to the CNN model.

### 3.4. Important Wavelengths

#### 3.4.1. Pre-Hoc Analysis

The results of the mutual information criterion, which is calculated before any model fit, are presented in Figure 11. This figure suggests that the exact peaks which indicate the most informative wavelengths are not unique across all spectral varieties. Indeed, for Malagouzia, the most important spectral region appears to be centered around 730 nm. For Chardonnay, two distinct spectral regions are at about 490 and 660 nm. The results from Sauvignon-Blanc also indicate the spectral region of 670 nm to be important, while two more peaks at 930 and 1890 nm also suggest that there are some regions in the upper NIR and SWIR that could be relevant. Finally, for the only red-grape variety, namely Syrah, the most important region is also at about 730 nm (like for the Malagouzia variety).

#### 3.4.2. Post-Hoc Analysis

In this section we present the results of the feature importances as ascribed by the best models for each learning algorithm, across all four grape varieties. These are presented in Figure 12 for PLS by plotting the VIP scores, Figure 13 for RF by visualizing the Gini score, and Figure 14 and Figure 15 for SVR and CNN, respectively, by plotting the SAGE values.

It is evident that PLS has produced feature rankings that are not entirely sparse, although some feature ranges and peaks may be identified even in this case. For Chardonnay, the VIP scores indicate that there are many areas which the model uses, and coupled with its low accuracy we can conclude that the model does not successfully model the relationship between spectrum and sugar content. With respect to Malagouzia, the most important appear to be centered around 670 nm, with lower importance given to 750, 1210, and 1920 nm. In regard to Sauvignon-Blanc, important peaks may be identified almost throughout the entire spectrum. The most significant of those are at 570, 710, 850, 1010, 1190, and 1870 nm. Similarly, in Syrah, there are multiple peaks with the areas centered around 800 and 1150 nm deemed to be the most significant.

For RF, the feature ranking produces much clearer results as there is greater sparsity and there are dominant peaks (Figure 13). In particular, for the Chardonnay variety, the most important wavelengths are 570, 610, 630, and 710 nm. It should be noted though, that for this variety, RF did not attain high accuracy. For Malagouzia, 570, 610, and 620 nm are ranked the highest, while for Sauvginon-Blanc 570, 720 and 1180 nm are the more pronounced. Finally, for Syrah, the most important wavelengths are 820 and 860 nm. Taking into consideration that RF attains the best results using the first derivative, the actual absorption bands may be slightly shifted in either direction of these identified bands.

The SAGE values for SVR are also quite sparse (Figure 14). For Chardonnay, the top-ranked features are at 570, 690, and at 2500 nm. This latter finding is interesting given that it lies at the very edge of the spectrum. Indeed, if Figure 7 is observed, there is a distinct difference in the spectral albedo starting from approximately 1100 nm until the edge of the spectrum, where it appears to be slightly more pronounced. As far as the Malagouzia variety is concerned, SVR regards as important the 570, 710, 960, 1000, and 1150 nm. A small (yet not essential) importance is ascribed also at 2470 nm. For Sauvignon-Blanc, SAGE indicates that 570, 710 and 2470 nm are the most significant; i.e., also using a band in SWIR to better model the input–output association. Finally, for Syrah, the most essential features are at 690 and 760 nm, while also the 2480 nm are noted, albeit with lower emphasis.

The SAGE values for the CNN model may also be characterized as sparse, with sharp peaks denoting crucial features (Figure 15). For Chardonnay, the three most important peaks are at 690, 930, and 2500 nm. The Malagouzia variety is modeled mostly via the 350, 580, 700, and 930 nm features. In Sauvignon-Blanc, where the CNN model attains the best accuracy and has the highest RPIQ difference from the other models, the important wavelengths are at 570, 700, 720, 960, and 1150 nm. Some noteworthy peaks are also at 670, 690, 820, and 2500 nm. Finally, the most decisive results are for the Syrah variety, where the area around 700 nm is considered highly important, with lower emphasis given in the visible part (350 to 380 nm, at the edge between ultraviolet and violet) and at 820 and 920 nm.

## 4. Discussion and Conclusions

The present study expands existing the approaches which aim to predict the sugar content of wine grapes to assess their maturity stage by using in situ spectral measurements. The spectral data were collected in the field using the contact probe of a spectrometer covering the VNIR–SWIR range (350 to 2500 nm), while four different varieties were covered. Following this, four learning algorithms were applied to predict the sugar content from the spectra, while accordingly nine spectral sources were considered (i.e., generated via spectral pre-processing). To assess the effectiveness of the models, a cross-validation technique ensured that the accuracy results are reported more robustly. Finally, a pre-hoc and a post-hoc interpretability analysis took place to identify the wavelengths and spectral regions deemed important by the best-performing models.

The accuracy results were promising for three varieties (Malagouzia, Sauvignon-Blanc, and Syrah) with an average $R^{2}>0.8$ and RPIQ>4 across the five folds for the best-performing models. In the fourth variety, namely Chardonnay, the results were acceptable with an average $R^{2}>0.6$ and RPIQ>2. The best results were attained using mostly first-derivative spectra (either from the raw reflectance or from the pseudo-absorbance transformation), while the best learning algorithm was the CNN for three out of four varieties with SVR slightly outperforming the CNN model for Chardonnay. The worse results of Chardonnay, when compared to the other varieties, may be attributed to the lower variance recorded in the physical sample (see, e.g., Figure 5), where there is a lack of samples with low sugar content.

Compared to other studies, our results are in accord with the literature in terms of accuracy of prediction. Arana et al. [12] examined two varieties (namely, Viura and Chardonnay) in the 800–2500 nm range, with the best results being in the Chardonnay variety with $R^{2}=0.70$. Excellent results were reported in the work of Larrain et al. [21], with an average $R^{2}$ of 0.90 across six varieties—where notably the worst performance was recorded in the Chardonnay variety ($R^{2}=0.87$)—while the spectral region analyzed was 640–1100 nm. The study of Beghi et al. [13] attained an $R^{2}$ of 0.62 for the Corvina cultivar using the 400–1000 nm spectral range and intact grapes. Daniels et al. [50] reported an $R^{2}$ of 0.76 for Sultana (also known as Thompson Seedless) table grapes, by recording whole bunches in the 800–2500 nm range in the laboratory. Using samples from across different grape varieties and stages of maturity, González-Caballero et al. [15] attained an $R^{2}$ of 0.9 while recording the 380–1700 nm spectral range.

In terms of feature importance, both pre- and post-hoc interpretability analysis identified mostly distinct wavelengths in the visible and near infrared. It should be noted that the dissolved solids in the aqueous solution of the grape juice, which are measured by the degrees of Brix, are primarily composed of glucose and fructose. The absorption bands due to O–H and C–H groups in those largely influence the spectral variation [51]. Moreover, another point that ought to be mentioned, is that the best models were developed using first-derivative spectra, therefore our identified wavelengths are located with a slight offset from the true absorption band. Coupled with the broad absorption bands even at the fundamental frequency [51], it is evident that no exact wavelength corresponding to a specific peak may be pointed out. Nevertheless, some wider areas have been identified by the best models. More concretely, the models in the present study mainly focus on 570–630 nm and 670–730 nm. The first region corresponds to the 5th overtone of O–H, while the latter to the 5th overtone of C–H and the [52]. On a smaller scale, the models identified the 800–860 nm region (possibly due to the second overtone of the O–H combination band [53]), 920–960 nm (overtones of O–H and C–H stretching bands), and the 1150–1200 nm region (overtones of O–H combinations and C–H stretching). Finally, the 2470–2500 nm band region identified by our models may be ascribed to C–H stretching and C–C and C–O–C stretching combinations found in polysaccharides (which produce constituent sugars) [52].

Compared to other studies, our findings regarding the most important wavelengths are similar. González-Caballero et al. [15] report the 728, 948, 976, 1138, and 1384 nm as important for TSS estimation across different wine grape varieties. In Ref. [23], it was determined that the most important wavelengths were 670, 730, and 780 nm for the Nebbiolo red wine grape variety. Kemps et al. [54] suggest that the 700 nm are important for three varieties (namely, Cabernet Sauvignon, Merlot, and Syrah) with the model for the Syrah variety also making use of the 1190 and 1400 nm. Another study that analyzed sucrose, glucose, and fructose [51] determined that the O–H bonds at 740, 770, 960, and 984 nm are correlated with sugar content, while C–H bonds at 910 and around 1015 are also significant. Similar results have also been reported for the sugar content determination of other fruits as well. For example, for the pomegranate fruit, the 780 and 950 nm have been identified [55]. For strawberry, the 580, 680, and 960 nm were shown to be important for TSS [56]. The apples’ sugar content was also determined via the 680 and 940 nm [57,58]. The above results are in accordance with our identified wavelengths, suggesting that our models have identified strong correlations between the input (diffuse reflectance spectra) and output (°Brix, or sugar content).

The current study is a proof of concept for crop assessment based on the exploitation of in situ spectral data and artificial intelligence methods to smartly capture information in real field conditions. Furthermore, it also takes into account the evaluation of different pre-processing and methods to estimate the feature importance, to further enhance the knowledge of the AI models’ capacity to capture different patterns in the spectral signatures, as highlighted by Silva et.al. [59]. This is just the beginning of what is expected to be a revolution in the way the agricultural robotic industry is operated. The grape spectral library, the insights from the feature selection and the novel CNN framework presented here can be the basis for automated in situ ripeness determination and selective harvesting utilizing edge computing techniques by (i) providing improvement in production leaving unripe products in the field to mature whilst identifying the ripe ones, and (ii) enabling some human-like activities to be performed by robots [60].

In the future, work can focus on using miniaturized NIR sensors that connect to mobile devices [9], or on hyperspectral cameras mounted on robotic platforms [61] having as a goal to automate the harvesting procedure [62]. More samples may also be recorded for the Chardonnay variety, particularly when the sugar content is low, to ascertain if the spectral model can improve in terms of prediction accuracy. Another option to enhance the prediction accuracy in the Chardonnay variety would be to utilize feature selection or dimensionality reduction techniques [63], including the simultaneous usage of multiple spectral pre-treatments and autoencoders [64] to compress the useful spectral information. Finally, the VNIR–SWIR spectra may be used to further estimate other maturity and oenological parameters, suchas the pH content and titratable acidity, to provide more robust ripeness estimation [17], potentially using multi-output CNNs.

## Figures and Tables

**Figure 1 sensors-23-01065-f001:**
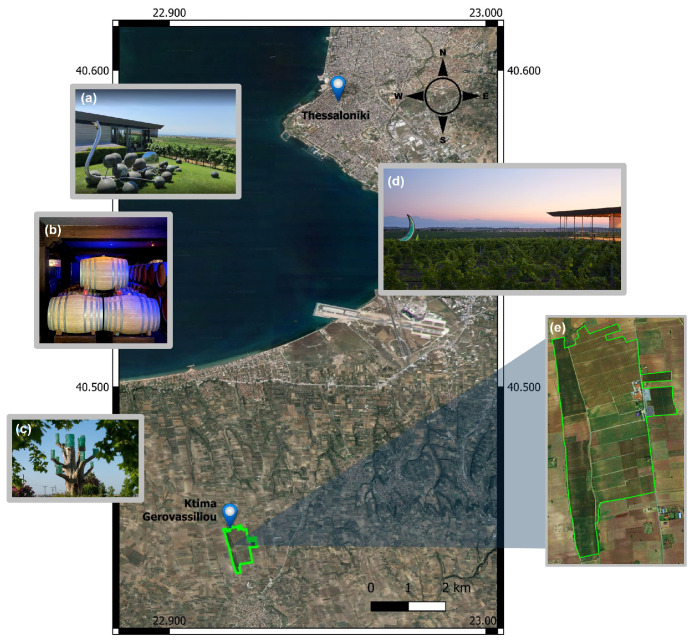
The pilot area of Ktima Gerovassiliou, located in Northern Greece. The sub figures (**a**–**d**) depict different views of the vineyard with its unique and special architecture while sub figure (**e**) presents the total cultivated area.

**Figure 2 sensors-23-01065-f002:**
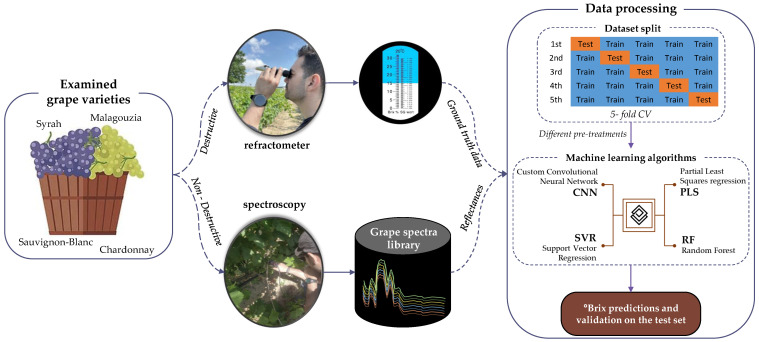
Overall experimental and methodological process adopted in the current work.

**Figure 3 sensors-23-01065-f003:**
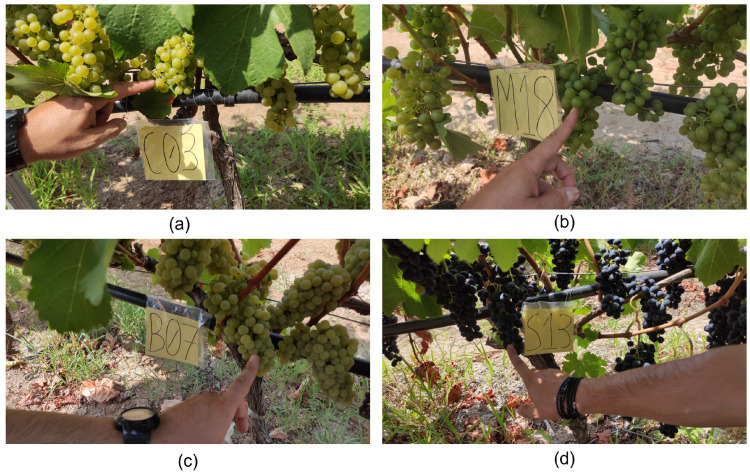
Labeled bunches in the four different examined varieties (**a**) Chardonnay, (**b**) Malagouzia, (**c**) Sauvignon-Blanc, and (**d**) Syrah.

**Figure 5 sensors-23-01065-f005:**
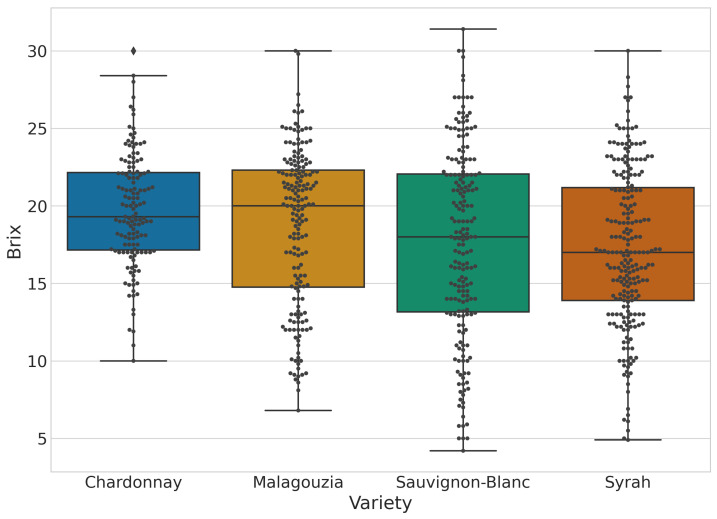
Box plot (overlaid with a swarmplot) of the °Brix content across the four different grape varieties; outlying values are denoted with a diamond shape.

**Figure 6 sensors-23-01065-f006:**
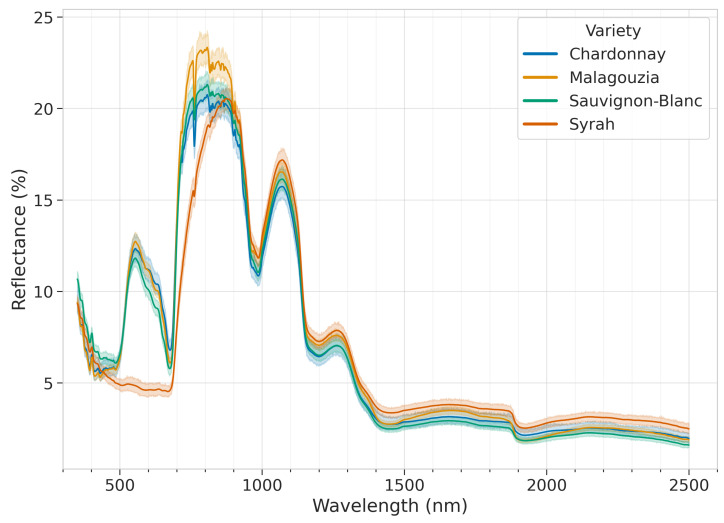
Mean spectrum (with confidence interval) of the reflectance spectra per each variety.

**Figure 7 sensors-23-01065-f007:**
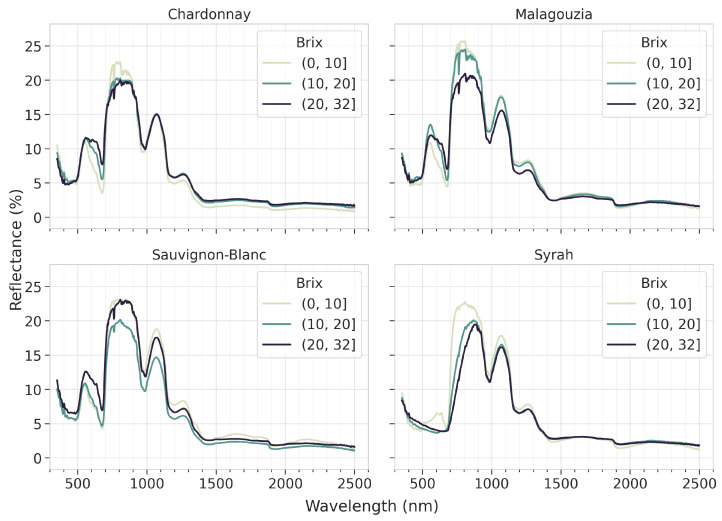
The effect of the change the °Brix content (grouped into three arbitrary classes) has on the mean reflectance spectra of each grape variety.

**Figure 8 sensors-23-01065-f008:**
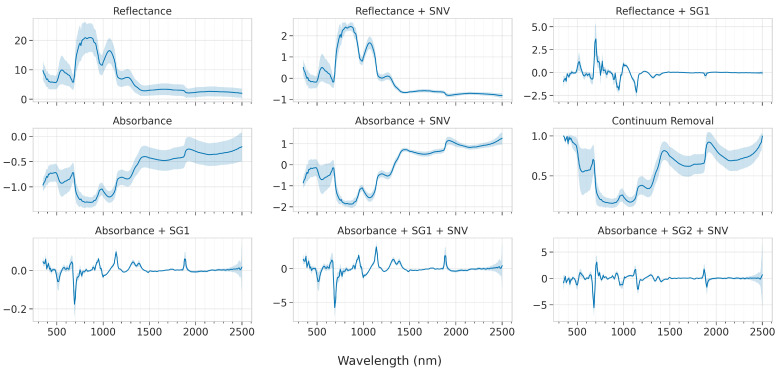
The effect of the various pre-treatments on the mean spectra (across all varieties); the shaded area indicates the standard deviation.

**Figure 9 sensors-23-01065-f009:**
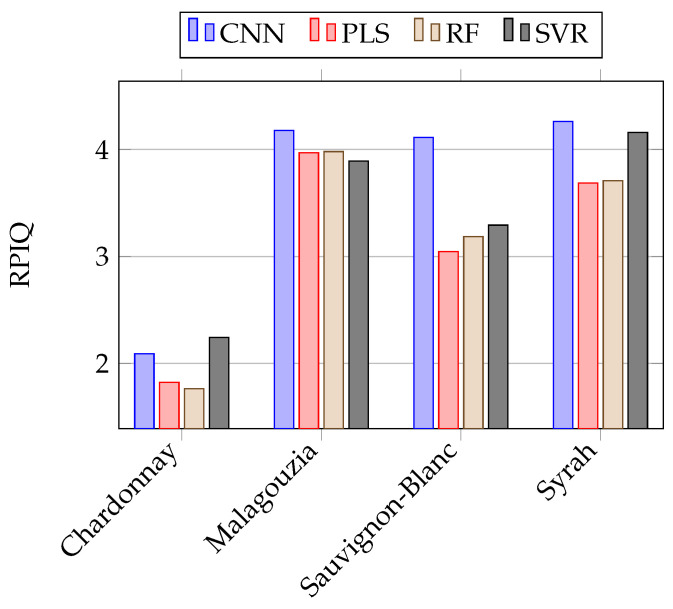
Visual comparison of the RPIQ metric (higher is better) of Table 3.

**Figure 10 sensors-23-01065-f010:**
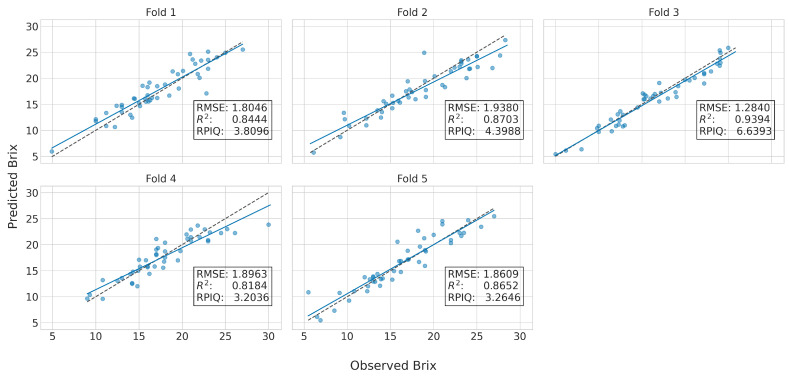
Predictions of the best CNN model in the Syrah variety across the five folds in the independent test set; the dashed line is the main diagonal whereas the continuous line is the least squares plot.

**Figure 11 sensors-23-01065-f011:**
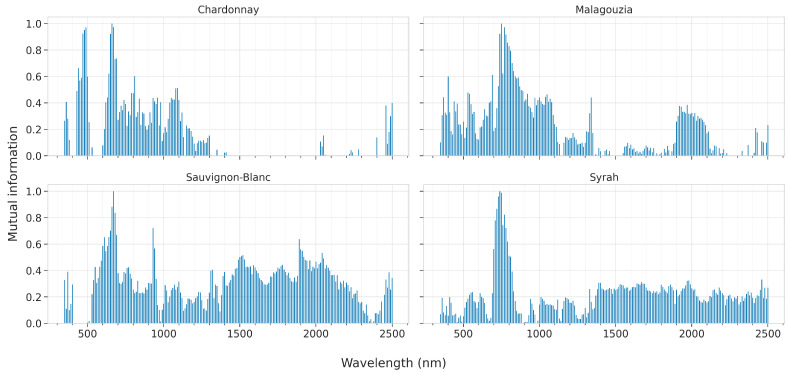
Mutual information-based feature ranking across the four grape varieties of the reflectance spectra; the higher the bar, the more informative the feature about the °Brix content.

**Figure 12 sensors-23-01065-f012:**
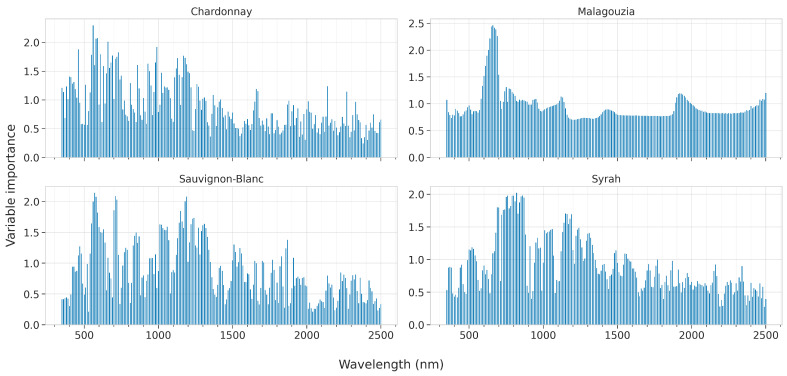
Mean feature importance according to Variable Importance in Projection values for the best PLS model across the five folds and for each of the four grape varieties; the higher the bar, the more informative the feature about the °Brix content.

**Figure 13 sensors-23-01065-f013:**
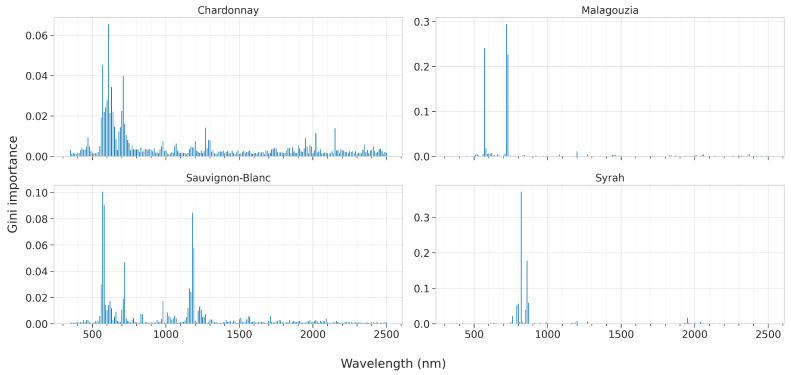
Mean feature importance according to Gini values (also known as impurity-based importances) for the best Random Forest model across the five folds and for each of the four grape varieties; the higher the bar, the more informative the feature about the °Brix content.

**Figure 14 sensors-23-01065-f014:**
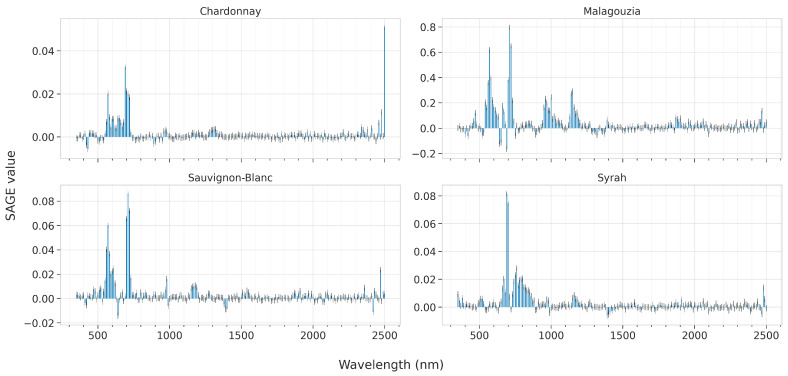
Mean feature importance according to SAGE values for the best SVR model across the five folds and for each of the four grape varieties; the higher the bar, the more informative the feature about the °Brix content.

**Figure 15 sensors-23-01065-f015:**
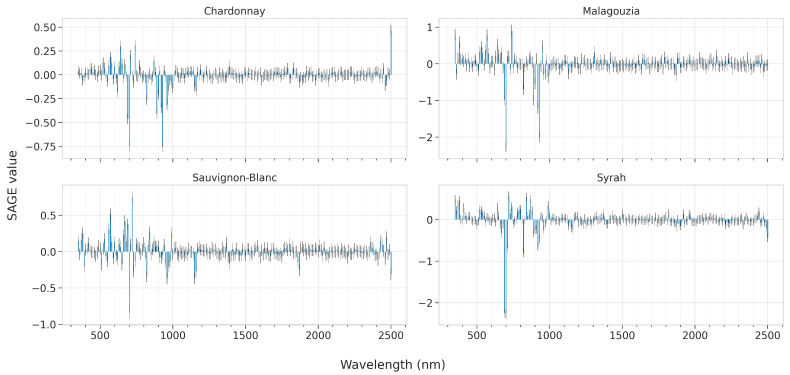
Mean feature importance according to SAGE values for the best CNN model across the five folds and for each of the four grape varieties; the higher the bar, the more informative the feature about the °Brix content.

**Table 1 sensors-23-01065-t001:** The hyperparameters of the CNN model optimized via the Hyperband algorithm.

Hyperparameter	Search Values
Input standardization	{none, min-max, standard-score}
No. of conv. layers	{1, 2, 3}
Conv. filters of layer	{8,10,…,32}
Kernel size	{3, 5, 7}
Batch normalization after conv. layer	{False, True}
Max pooling after conv. layer ^1^	{False, True}
No. of fully connected layers ^2^	{1, 2, 3}
No. of neurons in fully connected layer	{8,10,…,64}
Batch size	{4, 6, 8}
Learning rate of Adam optimizer	[$10^{-4},10^{-2}$] with log sampling

^1^ Using a max pooling window 2, i.e., halving. ^2^ Excluding the final inference layer.

**Table 2 sensors-23-01065-t002:** Major statistical moments of the °Brix content across the four different grape varieties that were collected, where Std is the standard deviation, Min and Max are the minimum and maximum values, and $Q_{1}$, $Q_{2}$ and $Q_{3}$ are the 1st, 2nd, and 3rd Quartile or 25th, 50th, and 75th percentiles, respectively.

Variety	Year	Samples per Year	Total Samples	Mean	Std.	Minimum	$Q_{1}$	$Q_{2}$	$Q_{3}$	Maximum
Chardonnay	2020	39	139	19.73	3.62	10.0	17.1	19.3	22.2	30.0
	2021	100								
Malagouzia	2020	0	179	18.63	5.06	6.8	14.7	20.0	22.3	30.0
	2021	179								

Sauvignon-Blanc	2020	95	195	17.61	6.14	4.2	13.1	18.0	22.1	31.4
	2021	100								

Syrah	2020	110	230	17.22	5.04	4.9	13.9	17.0	21.2	30.0
	2021	120								

**Table 3 sensors-23-01065-t003:** Results of the mean prediction error metrics on the independent test set across the 5-folds, for each grape variety and all models examined; for all models, the table also indicates which spectral pre-treatment yielded the best results, while boldface denotes the best results for each variety.

			Metric		
Variety	Model	Pre-Treatment	$R^{2}$	RMSE ^1^	RPIQ
Chardonnay	CNN	Ref+SG1	0.60	2.20	2.09
	PLS	Abs+SG2+SNV	0.46	2.54	1.82
	RF	Abs+SG1	0.46	2.58	1.76
	SVR	Abs+SG1	**0.63**	**2.10**	**2.24**

Malagouzia	CNN	Ref+SG1	**0.84**	**1.96**	**4.18**
	PLS	Abs	0.84	2.01	3.97
	RF	Ref+SG1	0.83	2.04	3.98
	SVR	Abs+SG1+SNV	0.83	2.07	3.89

Sauvignon-Blanc	CNN	Ref+SG1	**0.86**	**2.20**	**4.11**
	PLS	Abs+SG1+SNV	0.75	2.94	3.04
	RF	Abs+SG1+SNV	0.78	2.81	3.19
	SVR	Abs+SG1	0.80	2.70	3.29

Syrah	CNN	Ref+SG1	**0.87**	**1.76**	**4.26**
	PLS	Abs+SG1+SNV	0.84	1.98	3.69
	RF	Abs+SG1	0.83	1.99	3.71
	SVR	Abs+SG1	0.87	1.77	4.16

^1^ In °Brix.

## Data Availability

Not applicable.

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
