# Peer review of "Estimation of Sugar Content in Wine Grapes via In Situ VNIR–SWIR Point Spectroscopy Using Explainable Artificial Intelligence Techniques"

_sensors, 2023, doi:10.3390/s23031065_

Round 1

Reviewer 1 Report

Review report

As stated in the title, this study uses visible and near-infrared (VNIR) and short-wave infrared (SWIR) point spectroscopy in combination with artificial intelligence (AI) and deep learning techniques for the estimation of sugar content in wine grapes. There is no doubt that spectroscopic methods are versatile and data-rich, and AI techniques are of great assistance in acquiring meaningful insights.

As intriguing and promising as the title is, the content is equally well-designed and sufficient to support it. For the ease of understanding of the readers, the proposed protocol and study are well designed, explained, and discussed. The paper reports it in a comprehensive manner. In addition, I am also impressed by the quality of the English used in this article.

For improving the article, I was able to identify only a few very minor suggestions, which are listed below.

Abstract

- The abstract should briefly indicate from four different AI models which AI model is the most appropriate for such studies based on the studies presented in the article.

2.1.1 Study area

- In the caption of figure 1, please provide a description of the inset images.

2.2  Methods

- Figure 2, Data processing, is it correct that ‘Different pre-treatments’ have been applied after ‘Machine learning algorithms’?

2.2.1. Equipment, Preparation and Protocols

- Line 134, Please specify the range, accuracy, and resolution of the refractometer.

2.2.6 Model evaluation metrics

- N should be defined in equation 2 as well as Q3 and Q1 in equation 3.

4. Discussion and conclusions

- Is it possible to somehow improve the accuracy of results for Chardonnay, maybe using only selective features? Please comment.

General comment:

To overcome inherent biases of the investigator’s mind, such chemometric analysis must be done in a double-blind manner. Were these analyses done in such a way?

Reviewer 2 Report

Estimation of sugar content in wine grapes via in situ VNIR–SWIR point spectroscopy using explainable artificial intelligence techniques

Abstract

L4. Please use the past tense instead of the present.

L8. The authors could write the range of the phenological stages for which they collected data using the BBCH scale.

Introduction

L36-42. I recommend the authors include references to support their statements.

L74-86. I recommend the authors move these sentences to the Materials and Methods section.

I recommend the authors include one paragraph explaining the importance of using machine learning techniques in the wine sector by providing some examples.

Materials and Methods

Study Area

I recommend the authors include more information related to the study area (e.g., vine varieties used in the study, year of establishment, rootstock, trellis system, type of agricultural system (conventional or organic), water application (rainfed or drip irrigation)).

Methods

L101. I recommend the authors remove this from the sentence “For the convenience of the reader,”.

Grape spectra library development – field sampling

I recommend the authors include the phenological stages for which they took samples using the BBCH scale.

I recommend the authors put the figures and tables near their cross-references in the text.

Results and Discussion

I recommend the authors put the figures and tables near their cross-references in the text.

I recommend the authors remove “Discussion” from the section title since they refer only to the results in this section.

Appendix A

This section should include Table A1 and not the References section.

General Comment

I recommend that this manuscript be accepted after minor revisionS. The study assesses different machine learning methods that use spectral DATA to estimate the sugar content of wine varieties. The described methodology can contribute to the fast and non-destructive estimation of important berry parameters and promote optimal grape harvesting and, consequently, less food waste.   
